# Identification of Novel Mitochondrial Pyruvate Carrier Inhibitors by Homology Modeling and Pharmacophore-Based Virtual Screening

**DOI:** 10.3390/biomedicines10020365

**Published:** 2022-02-02

**Authors:** Lamees Hegazy, Lauren E. Gill, Kelly D. Pyles, Christopher Kaiho, Sophia Kchouk, Brian N. Finck, Kyle S. McCommis, Bahaa Elgendy

**Affiliations:** 1Center for Clinical Pharmacology, Washington University School of Medicine, University of Health Sciences & Pharmacy, St. Louis, MO 63110, USA; Christopher.kaiho@uhsp.edu (C.K.); sophia.kchouk@uhsp.edu (S.K.); belgendy@wustl.edu (B.E.); 2Department of Pharmaceutical and Administrative Sciences, University of Health Sciences & Pharmacy, St. Louis, MO 63110, USA; 3Department of Biochemistry & Molecular Biology, Saint Louis University School of Medicine, St. Louis, MO 63104, USA; lauren.gill@slu.edu (L.E.G.); kelly.pyles@health.slu.edu (K.D.P.); 4Department of Medicine, Washington University School of Medicine, St. Louis, MO 63110, USA; bfinck@wustl.edu; 5Chemistry Department, Faculty of Science, Benha University, Benha 13518, Egypt

**Keywords:** mitochondrial pyruvate carrier, homology modeling, mutagenesis, pharmacophore modeling, virtual screening

## Abstract

The mitochondrial pyruvate carrier (MPC) is an inner-mitochondrial membrane protein complex that has emerged as a drug target for treating a variety of human conditions. A heterodimer of two proteins, MPC1 and MPC2, comprises the functional MPC complex in higher organisms; however, the structure of this complex, including the critical residues that mediate binding of pyruvate and inhibitors, remain to be determined. Using homology modeling, we identified a putative substrate-binding cavity in the MPC dimer. Three amino acid residues (Phe66 (MPC1) and Asn100 and Lys49 (MPC2)) were validated by mutagenesis experiments to be important for substrate and inhibitor binding. Using this information, we developed a pharmacophore model and then performed a virtual screen of a chemical library. We identified five new non-indole MPC inhibitors, four with IC_50_ values in the nanomolar range that were up to 7-fold more potent than the canonical inhibitor UK-5099. These novel compounds possess drug-like properties and complied with Lipinski’s Rule of Five. They are predicted to have good aqueous solubility, oral bioavailability, and metabolic stability. Collectively, these studies provide important information about the structure-function relationships of the MPC complex and for future drug discovery efforts targeting the MPC.

## 1. Introduction

Mitochondria convert chemical energy stored in nutrients to adenosine triphosphate (ATP) and other forms of energy and play a number of important roles in intermediary metabolism. Pyruvate is used by the mitochondrion for oxidative metabolism by pyruvate dehydrogenase and is also carboxylated to provide anaplerotic substrates for the tricarboxylic acid (TCA) cycle. Transport of pyruvate across the inner mitochondrial membrane into the mitochondrial matrix is required for oxidation or carboxylation and is mediated by the mitochondrial pyruvate carrier (MPC) [1,2]. The MPC is a two-subunit carrier complex composed of MPC1 and MPC2 proteins and localized to the mitochondrial inner membrane (MIM) that works as a gatekeeper for pyruvate entry into mitochondria. These two subunits are paralogous proteins encoded by two genes that are highly conserved from yeast to humans. It is generally believed that MPC1 and MPC2 form a functional heterodimer; it has been shown that deletion of either MPC subunit leads to the destabilization and degradation of the other MPC protein [3,4]. Global genetic deletion of either MPC gene in mice is not compatible with life [3,4], likely due to developmental defects in the central nervous system and heart. However, conditional deletion in many tissues is well tolerated and can actually protect from the development of metabolic disease [5,6,7].

Development of specific and potent MPC modulators holds a great potential as therapeutics for myriad diseases including type 2 diabetes, nonalcoholic steatohepatitis (NASH), cancer, and even hair loss [8,9,10,11]. Known MPC inhibitors fall mainly into two categories: i. α-cyanocinnamate derivatives (e.g., UK-5099) [12]; and ii. thiazolidinediones (TZDs) (e.g., rosiglitazone) [13], which lack selectivity, as most TZDs bind and activate the peroxisome proliferator-activated receptor-gamma (PPARγ) (Figure 1). Compounds GW604714X and GW450863X are two potent thiazolidine inhibitors, which are comparable to UK-5099 in potency, but are also known to affect the activity of K_ATP_ channels to inhibit potassium import [14].

Despite widespread interest, development of specific and targeted MPC inhibitors is difficult due to the lack of structural information about the MPC complex. Additionally, lack of specific chemical probes for MPC hinders the ability to study the relationship between MPC phenotypic data and to launch the clinical translatability of this target. Herein, we identify key amino acids required for the binding of MPC inhibitors and report the identification of novel MPC inhibitors using a ligand-based pharmacophore modeling approach.

## 2. Materials and Methods

### 2.1. Chemistry

All starting materials were purchased from commercial suppliers and used without further purification. The purities of the final compounds were characterized by high-performance liquid chromatography (LC/MS) using a gradient elution program (Ascentis Express Peptide C18 column, acetonitrile/water 5/95/95/5, 5 min, 0.05% trifluoracetic acid) and UV detection (254 nM). The purities of final compounds were 95% or greater. ^1^H NMR spectra was recorded on Varian 500 MHz operating at 500 MHz for ^1^H NMR and ^13^C NMR spectra was recorded on a Bruker NMR 400 MHz Avance III spectrometer operating at 100 MHz for ^13^C NMR. Chemical shifts are given in part per million (ppm) relative to the deuterated solvent residual peak, coupling constants *J* are given in Hertz.

#### 2.1.1. General Procedure for the Synthesis of 3-Substituted-2-Cyano-2-Propenoic Acids

Step 1: Synthesis of 3-substituted-2-cyano-2-propenoic acid ethyl ester

To a solution of aldehyde (1 equiv.) and ethylcyanoacetate (1.3 equiv.) in ethanol (3 mL) was added piperidine (0.2 quiv.) and the reaction mixture was stirred at 60 °C for 12 h. The desired product crystallized out of solution upon cooling and was filtered and washed with H_2_O (2x), a mixture of EtOAc/Hexanes (1:4) (2x) and dried under vacuum.

Step 2: Synthesis of 3-substituted-2-cyano-2-propenoic acids

Ethyl cycanoacrylate derivatives (0.5 mmol) were dissolved in a 1:1 mixture of 0.5 N LiOH solution and THF overnight at room temperature. The THF solvent was removed under reduced pressure and a 2 N solution of HCl was added and the solution was stirred for 1 h at room temperature, resulting in the precipitation of white crystals. The resulting crystals were filtered with H_2_O (2x) and a mixture of EtOAc/Hexanes (1:4) (2x) and were dried under vacuum to yield the final product in good yields.

#### 2.1.2. (E)-3-[5-(1,3-Benzothiazol-2-yl)-2-Furyl]-2-Cyanoacrylic Acid (BE1976)

Yellow solid (87%). ^1^H NMR (500 MHz, acetone-*d6*) *δ* 98.25 (dd, *J* = 8.2, 1.2 Hz, 1H), 8.17 (s, 1H), 8.10 (d, *J* = 8.2 Hz, 1H), 7.64 (q, *J* = 3.8 Hz, 2H), 7.60 (ddd, *J* = 8.3, 7.1, 1.3 Hz, 1H), 7.52 (ddd, *J* = 8.3, 7.2, 1.2 Hz, 1H); ^13^C NMR (100 MHz, DMSO-*d6*) *δ* 164.04, 156.26, 154.10, 152.64, 150.29, 138.20, 135.34, 126.95, 126.27, 116.31, 114.70, 101.07. ESI-MS (*m*/*z*): 297 (M + 1)^+^.

#### 2.1.3. (E)-2-Cyano-3-(1-Phenyl-4-Pyrazolyl)acrylic Acid (BE1978)

White solid (52%). ^1^H NMR (500 MHz, acetone-*d6*) *δ* 9.06 (d, *J* = 2.2 Hz, 1H), 8.54 (d, *J* = 2.2 Hz, 1H), 8.40 (d, *J* = 2.2 Hz, 1H), 7.95 (dd, *J* = 8.2, 2.3 Hz, 2H), 7.63 (td, *J* = 8.6, 8.1, 2.2 Hz, 2H), 7.48 (dd, *J* = 8.6, 6.5 Hz, 1H); ^13^C NMR (100 MHz, DMSO-*d6*) *δ* 163.91, 145.99, 141.81, 138.95, 132.86, 130.27, 128.27, 119.73, 117.90, 117.26, 100.80. ESI-MS (*m*/*z*): 240 (M + 1)^+^.

#### 2.1.4. (E)-2-Cyano-3-(3,5-Dimethyl-1-Phenyl-4-Pyrazolyl)acrylic Acid (BE1980)

White solid (74%). ^1^H NMR (500 MHz, DMSO-*d6*) *δ* 9.06 (d, *J* = 2.2 Hz, 1H), 8.54 (d, *J* = 2.2 Hz, 1H), 8.40 (d, *J* = 2.2 Hz, 1H), 7.95 (dd, *J* = 8.2, 2.3 Hz, 2H), 7.63 (td, *J* = 8.6, 8.1, 2.2 Hz, 2H), 7.48 (dd, *J* = 8.6, 6.5 Hz, 1H) 8.22 (s, 1H), 7.62–7.42 (m, 5H), 2.38 (s, 3H), 2.35 (s, 3H); ^13^C NMR (100 MHz, DMSO-*d6*) *δ* 164.07, 149.65, 147.90, 142.57, 138.70, 129.82, 128.90, 125.33, 117.04, 114.30, 103.28, 14.09, 14.05. ESI-MS (*m*/*z*): 268 (M + 1)^+^.

#### 2.1.5. (E)-2-Cyano-3-(p-Tolyl)acrylic Acid (BE1984)

White solid (33%). ^1^H NMR (500 MHz, acetone-*d6*) δ 8.34 (s, 1H), 8.09–7.98 (m, 2H), 7.47 (d, *J* = 8.1 Hz, 2H), 2.48 (s, 3H); ^13^C NMR (100 MHz, DMSO-*d6*) *δ* 163.72, 154.62, 144.30, 130.24, 130.19, 116.56, 102.64, 21.64, 21.59. ESI-MS (*m*/*z*):186 (M + 1)^+^.

#### 2.1.6. (E)-2-Cyano-3-(4-Phenyl-3-Pyrazolyl)acrylic Acid (BE 2617)

Light brown solid (32%). ^1^H NMR (500 MHz, DMSO-*d6*) δ 8.58 (s, 1H), 8.03 (d, *J* = 3.6 Hz, 1H), 7.72–7.33 (m, 5H); ^13^C NMR (100 MHz, DMSO-*d6*) δ 164.20, 146.14, 129.59, 129.44, 117.39, 112.56, 99.53. ESI-MS (*m*/*z*): 240 (M + 1)^+^.

#### 2.1.7. (E)-2-Cyano-3-[4-(p-Tolyl)-3-Pyrazolyl]acrylic Acid (BE 2623)

White solid (57%). ^1^H NMR (500 MHz, DMSO-*d6*) δ 9.87 (d, *J* = 3.2 Hz, 1H), 8.01 (d, *J* = 3.0 Hz, 1H), 7.74–7.63 (m, 2H), 7.51–7.18 (m, 5H), 2.36 (d, *J* = 2.9 Hz, 3H). ESI-MS (*m*/*z*): 254 (M + 1)^+^.

### 2.2. Homology Modeling

MPC1 and MPC2 monomers were modeled using the I-TASSER server based on the X-ray structures of bacterial SemiSWEET crystal structures [15,16,17]. I-TASSER uses a hierarchical approach to protein structure prediction and structure-based function annotation [18,19,20]. It first identifies structural templates from the PDB by the multiple threading approach, with full-length atomic models constructed by iterative template-based fragment assembly simulations. It was ranked as the top server for automated protein 3D structure prediction in recent CASP community-wide experiments for eight consecutive times. A model of the MPC dimer binding site was constructed by structural aliment using UCSF Chimera [21] based on the SemiSWEET transporter X-ray structure (PDB: 4X5M) as a template [16].

### 2.3. Pharmacophore Modeling, Database Preparation, and Pharmacophore-Based Virtual Screening

To develop a predictive model for a set of MPC inhibitors that can be used in virtual screening, we developed a pharmacophore model based on the chemical and geometrical features of UK-5099 using Phase [22]. The generated pharmacophore hypothesis is composed of three features: One negative charge (N) corresponding to the carboxylic group, one hydrogen bond acceptor (A) corresponding to the cyano group and one aromatic ring (Ar) corresponding to the indole ring. We then performed pharmacophore-based virtual screening of one million compounds from Enamine screening collection libraries to identify compounds that match the geometrical and chemical features represented by the pharmacophore hypothesis. The database was first processed by generating different protonation states, stereochemistry, tautomers, and ring conformations [23]. The relevant conformers (hits) were retrieved and aligned to the hypothesis. The hits were ranked by their PhaseScreenScore (fitness score), which is a score that measures how well the matching vector features (acceptors, donors, aromatic rings) overlay those of the hypothesis, and how well the matching conformation superimposes, in an overall sense, with the reference ligand conformation [24]. The fitness score was defined by:S = W_site_ (1 − S_align_/C_align_) + W_vec_ S_vec_ + W_vol_ S_vol_ + W_ivol_ S_ivol_(1)

S_align_ was the alignment score: RMS deviation between the site point positions in the matching conformation and the site point positions in the hypothesis. All user-adjustable parameters were kept as default values. C_align_; alignment cutoff, default value = 1.2. W_site_; Weight of site score, default value = 1.0. S_vec_ Vector score; average cosine between vector features in the matching conformation and the vector features in the reference conformation. W_vec_ Weight of vector score, default value 1.0. S_vol_ Volume score: Ratio of the common volume occupied by the matching conformer and the reference conformer, to the total volume (the volume occupied by both). Volumes were computed using van der Waals models of all non-hydrogen atoms. W_vol_ Weight of volume score, default value = 1.0. S_ivol_ Included volume score: Ratio of the volume overlap between the matching conformer and the included volumes (if present) to the total included volume. Volumes were computed using van der Waals models of all atoms, except nonpolar hydrogens. W_ivol_ Weight of volume score, default value = 0.0.

### 2.4. Bioluminescent Resonance Energy Transfer (BRET)-Based Assays for Inhibitor Binding

Based on the success of a previously described MPC BRET reporter assay [25], we developed our own BRET assay using more recently adopted BRET donor and acceptor pairs. In this assay, Nanoluciferase is used as the BRET donor, and mCherry fluorescent protein is used as the BRET acceptor. Human *MPC1* and *MPC2* cDNA constructs with stop codon removed for C-terminal tagging were obtained from the Harvard PlasmID repository. MPC2 was cloned into pcDNA3.1-cccdB-Nanoluc obtained from Addgene (87067) by Gateway cloning; MPC1 was cloned into pDest-mCherry-N1, obtained from Addgene (31907), also by Gateway cloning. Site-directed mutagenesis (Agilent) was then performed to delete linking regions between the MPC proteins and C-terminal tags.

For BRET assays, U2OS cells were transiently transfected with MPC1-mCherry and MPC2-NLuc constructs and, 24 h later, plated onto 9 columns of a clear-bottom white 96-well plate at 10,000 cells/well. As a BRET control, some cells were transfected with MPC2-NLuc only, and these cells plated on the remaining 3 columns of the 96-well plate at 10,000 cells/well. The following day, cells were washed with PBS and starved in phenol red-, glucose-, and pyruvate-free Dulbecco’s Modified Eagle’s Medium (DMEM) for 3 h. Cells were then washed in PBS and then provided PBS with 1 mM CaCl_2_ and 0.5 mM MgCl_2_ plus 2 μg/mL coelenterazine h BRET substrate, a white sticker applied to the bottom of the plate, and the plate immediately placed into a BioTek Synergy plate reader. Luminescence was then read repeatedly from the top of the plate at emission wavelengths of 485/20 nm and 640/40 nm for NLuc and mCherry signals, respectively. After several baseline readings, the plate was ejected, and test substrates/compounds were manually pipetted into the wells. All assays contained dimethyl sulfoxide (DMSO) vehicle-treated cells, as well as 5 μM UK-5099 positive control cells. Pyruvate was treated at 5 mM, and all other test compounds were dosed at 10 μM. To calculate BRET activity, at each reading interval, the luminescence of 620 nm emission wavelength was divided by the 485 nm emission wavelength, and the values from the MPC2-Nluc-only expressing cells were subtracted from the cells expressing both constructs. All data was then normalized to the values immediately prior to compound injection. Data were either reported as curves over time or average normalized BRET of the several readings immediately after compound injections. For testing pyruvate/inhibitor-binding residues, point mutations were made in MPC1 or MPC2 constructs by site-directed mutagenesis (Agilent); BRET assays were performed the same as above. All of these assays contained 96-well plates directly comparing cells expressing WT MPC and mutant BRET constructs.

### 2.5. Experimental Animals

Care and use of C57Bl/6J mice conformed to standards established by the National Institutes of Health, approved by the IACUC of Saint Louis University #2845. Mice were housed, up to 5 mice per cage, in ventilated cages and provided ad libitum access to standard rodent chow (5053 LabDiet) and water. Temperature and humidity of animal rooms was continuously monitored; rooms were set to a 12 h light/dark cycle at 6:00 AM–PM. Male and female mice were used for the described ex vivo experiments, at an age range of 8–20 weeks of age. For mitochondria isolation, mice were euthanized by CO_2_ asphyxiation. For primary hepatocyte isolation, mice were anesthetized with 2% isoflurane in 100% oxygen, and then euthanized by cervical dislocation. Additional mitochondrial respiration experiments were performed in mitochondria isolated from cardiac-specific MPC2−/− mice, which were generated and described previously [26].

### 2.6. Mitochondrial Isolation and Respiration

Hearts were removed from mice after CO_2_ asphyxiation and homogenized in buffer containing 250 mM sucrose, 10 mM Tris base, and 1 mM EDTA (pH = 7.4) by 8–10 passes of a glass-on-glass Dounce homogenizer on ice. Homogenates were centrifuged at 1000× *g* for 5 min at 4 °C to pellet nuclei and undisrupted cells. The supernatants were then centrifuged at 10,000× *g* for 10 min at 4 °C to enrich for mitochondria; this mitochondrial pellet was washed and re-pelleted twice in fresh sucrose/tris buffer. The mitochondrial pellet was then solubilized in ~150 μL of Mir05 respiration buffer (0.5 mM EGTA, 3 mM MgCl_2_, 60 mM lactobionic acid, 20 mM taurine, 10 mM KH_2_PO_4_, 20 mM HEPES, 110 mM sucrose and 1 g/L of fatty acid free bovine serum albumin; pH 7.1). Mitochondrial protein content was then measured by BCA; 50 µg of mitochondrial protein was added to the chambers of an Oxygraph O2K (Oroboros Instruments), with a total volume of 2 mL Mir05 buffer set to 37 °C. Respiration was stimulated with 5 mM pyruvate/2 mM malate and 2 mM ADP. After obtaining steady state respiration measurements, compounds were added to the chamber at the indicated concentrations. Then, 5 mM succinate was added to determine inhibitor specificity towards pyruvate-stimulated respiration. Steady-state rates of oxygen consumption were assessed for 1–2 min before addition of subsequent substrate or inhibitor. Oxygen consumption rates (OCR) were calculated from the change in oxygen concentration over time and normalized to 50 µg of mitochondria within the chamber. For the experiments shown, respiratory control ratios (RCRs) were ~7–10, indicating high quality mitochondrial preparations.

### 2.7. Primary Murine Hepatocyte Isolation and Culture

Primary murine hepatocytes were isolated by perfusion of the liver with 1 mg/mL collagenase through the portal vein as previously described [5]. Cells were counted and plated onto 6-cm collagen-coated dishes in high glucose DMEM + 10% fetal bovine serum (FBS), Pen/Strep, and amphotericin B overnight. The following day, cells were treated with media + DMSO vehicle or 10 μM MPC inhibitors for 20 min. Cells were then scraped and collected into protein lysate buffer (15 mM NaCl, 25 mM Tris Base, 1 mM EDTA, 0.2% NP-40, 10% glycerol) with 1× protease inhibitor cocktail (cOmplete, Roche) and phosphatase inhibitors (1 mM Na_3_VO_4_, 1 mM NaF, and 1 mM PMSF).

### 2.8. Western Blotting Procedures

Protein concentrations were determined using a Pierce MicroBCA kit (ThemoFisher), and 40 μg of protein was loaded into the lanes of precast criterion 4–15% tris-glycine gels (Bio-Rad) and electrophoresed. Proteins were then transferred to Immobilon PVDF membranes (MilliporeSigma). Membranes were blocked with 5% BSA in TBS-T for at least 1 h at room temperature. Primary antibodies were used at 1:1000 dilution in 5% BSA-TBS-T overnight; then, membranes were washed 3× in TBS-T and subjected to LiCor near-IR secondary antibodies at 1:10,000 dilution in 5% BSA-TBS-T for 1 h. After washing 3× in TBS-T, membranes were developed using a LiCor Odyssey imaging system. Antibodies used were anti-mCherry (Cell Signaling 43590S), anti-MPC2 (Cell Signaling 46141S), anti-phosphorylated PDH E1α S232, S293, and S300 (MilliporeSigma AP1062, AP1063, AP1064), anti-PDH Cocktail (Abcam ab110416), and anti-αTubulin (Sigma T5168); secondary antibodies used were anti-rabbit or anti-mouse IRDYE donkey 800CW (LiCor 926-32213 or 92632212). Full, un-cropped western blot images are provided in Appendix A.

### 2.9. Statistical Analyses

All data are presented as mean ± standard error of the mean, with statistical significance defined as *p* < 0.05. Data were analyzed by one- or two-way ANOVA as appropriate, using GraphPad Prism version 9.3.1. Post hoc analysis was performed using Tukey’s multiple comparison tests.

## 3. Results

### 3.1. Homology Modeling of the MPC and Mutagenesis Studies

The hMPC1/hMPC2 heterodimer has a greater binding efficiency for substrate and inhibitors than homodimers; the physiological relevance of MPC homodimers is not clear, due to the lack of stability of homodimers composed of endogenous proteins [1,2,3,4,5,6,7]. In order to map the substrate binding site and to gain understanding of the amino acid residues involved in substrate binding, we performed homology modeling of the heterodimer of MPC1 and MPC2 using I-TASSER (Iterative Threading Assembly Refinement) [18,19,20]. The server reported the AtSWEET13 sugar transporter (PDB:5XPD) to have the closest structural similarity for both monomers [15]. The dimer was constructed in the inward facing position based on structural alignment with the SemiSWEET transporter X-ray structure (PDB: 4X5M) as a template [16] (Figure 2A,B). During our work on this manuscript, Xu, et al., published a homology model of the MPC dimer and a molecular dynamics simulations study on the model [27]. Comparison of our homology model with the published model revealed high similarity. We identified potential substrate binding site residues and validated them using mutagenesis experiments. Structural alignments with other homolog transporters (SemiSWEET transporter PDB: 4QND and AtSWEET13 sugar transporter PDB: 5XPD) predicted the location of the substrate binding cavity as the central cavity formed between both monomers (Figure 2C) [16,17].

To investigate the functional role of this putative substrate-binding pocket, we used site-directed mutagenesis to change MPC1 residues Asn33, His84 and Phe66 and MPC2 residues Trp82, Lys49 and Asn100 individually to alanine and assessed the effects of each of these mutations on the interaction with pyruvate or known MPC inhibitors using a bioluminescence resonance energy transfer (BRET)-based MPC reporter assay (Figure 3). In this system, MPC2 protein is C-terminally fused to Nanoluciferase (NLuc; photon donor), while MPC1 is C-terminally fused to mCherry (BRET acceptor). Pyruvate and MPC inhibitors induce a conformational change in the complex, reducing the distance between the donor and the acceptor, resulting in an increase in BRET signal. Mutation of MPC1 F66A, MPC2 K49A, or MPC2 N100A decreased the ability of pyruvate or inhibitors to increase the BRET signal, suggesting that the hydrophilic moiety of MPC2 Lys49 and Asn100 side chains and the hydrophobic aromatic side chain of MPC1 Phe66 are important for substrate or inhibitor binding (Figure 3A). This loss of BRET activity was not due to changes in MPC expression of these mutant constructs, as all mutants were well expressed (Figure 3B). These data further suggest that inhibitor and substrate binding to the MPC dimer involves hydrophobic or π-π stacking interaction with the aromatic side chain of Phe66 and a salt bridge or hydrogen bonding interaction with the side chains of Lys49 and Asn100.

### 3.2. Pharmacophore Modeling

To further identify inhibitor features responsible for ligand binding and to identify novel inhibitors, we developed a 3D-ligand-based pharmacophore hypothesis using Phase Suite [24]. These methods identify compounds that retain or match known features (acceptor, donor, negative ionic, positive ionic, hydrophobic, and aromatic ring) using known actives as a training set. Phase software generates pharmacophore hypotheses using a common pharmacophore perception algorithm based on conformational alignment of pharmacophore features. The hypothesis was built based on the structure of **UK-5099**, and is composed of three features: one negative charge (N) corresponding to the carboxylic group, one hydrogen bond acceptor (A) corresponding to the cyano group and one aromatic ring (Ar) corresponding to the indole ring (Figure 4). This model is consistent with the homology modeling and mutagenesis data, which suggested the presence of π-π stacking and two electrostatic interactions as important interactions for MPC inhibitor binding. Following the model generation, we conducted a pharmacophore-based virtual screen of over one million compounds from the Enamine advanced collection library. The screening collection was prepared and screened as discussed in the experimental section. To be classified as a hit, a compound was required to match all pharmacophore features of the pharmacophore hypothesis. The scoring function, PhaseScreenScore (Fitness score), was used to rank the screened compounds. The PhaseScreenScore measures how well the conformer matches the hypothesis and is a linear combination of the volume, site, and vector alignment scores. The top 7 scored compounds based on the PhaseScreenScore were selected for biological testing (Figure 5 & Table 1). Two additional compounds (**BE2617**, **BE2623**) were synthesized based on structural similarity.

### 3.3. Identification and Validation of Novel MPC Inhibitors

To validate the ability of identified compounds to inhibit MPC, we again used the BRET-based system that is sensitive to pharmacologic inhibitors of the MPC. We were delighted to find that five compounds (**BE1976**, **BE1978**, **BE1980**, **BE1984**, and **BE1985**) increased BRET activity, indicating a direct interaction (Figure 6A). The identification of these new non-indole inhibitors increases the chemical space around this scaffold and provides opportunity for drug design. Four of the five active compounds were synthesized in-house for further confirmation (see experimental section). Knoevenagel condensation of the appropriate aldehyde (**1a–f**) with 2-cyanoacetate (**2**) to form arylidenecyanoacetic acid ethyl ester (**3a–f**) followed by saponification using LiOH leads to the formation of **BE1976**, **BE1978**, **BE1980,** and **BE1984** (Figure 1). Two more derivatives **BE2617** and **BE2623** were synthesized and showed activity in the BRET assay. These two compounds were synthesized to explore the activity of 1*H*-pyrazol-3-yl as a core heterocycle vs. 1*H*-pyrazol-3-yl in **BE1978**, **BE1980**, and **BE1985**.

Among the novel hits, **BE1976** (2,5-substituted furan) and **BE1978** (N-1 substituted pyrazole) showed the highest potency with an IC_50_ of 33 nM and 117 nM, respectively, for inhibiting mitochondrial pyruvate respiration (Table 1). **BE1976** was >7-fold more potent than **UK-5099**, while **BE1978** was >2-fold more potent than **UK-5099** (Table 1). The 3,5-dimethyl-1-phenyl pyrazole derivative **BE1980** (IC_50_ = 162 nM) was a potent inhibitor but was weaker than **BE1978**, which showed that substitution at 3,5 positions of the pyrazole ring were tolerated but not favored. 3-Bromophenyl-1*H*-pyrazol cyanoacrylic acid **BE1985** showed good inhibitory activity with an IC_50_ of 0.638 µM. **BE1985** was 5- and 4-fold less potent than the two corresponding N-1-substituted analogs **BE1978** and **BE1980**, respectively (Table 1). The *p*-tolyl cyanoacrylic acid **BE1984** showed weaker inhibition (IC_50_ = 1.533 µM) than the other heterocyclic analogs. Moreover, these compounds also inhibited mitochondrial respiration in a dose-dependent manner when pyruvate was provided as the respiratory substrate (Figure 6B). Additionally, these compounds did not reduce pyruvate respiration in MPC2−/− cardiac mitochondria (Figure 6C) indicating MPC-dependent inhibition. Lastly, these compounds also did not increase the phosphorylation of pyruvate dehydrogenase (PDH) E1α, a proxy for decreased PDH activity, suggesting that these compounds decrease pyruvate oxidation by MPC inhibition and not by reduction of PDH activity (Figure 6D).

The two synthesized 1*H*-pyrazol-3-yl derivatives **BE2617** and **BE2623** were also found to be MPC inhibitors. Compound **BE2617** was one of the most potent MPC inhibitors identified in this study, with an IC_50_ of 39 nM. **BE2617** was comparable to **BE1976** and was 6-fold more potent than **UK-5099**. The second derivative **BE2623** (IC_50_ = 0.731 µM) was more than 18-fold less potent than **BE2617**. Interestingly, the minor structural difference between **BE2617** and **BE2623** resulted in a significant difference in activity, which warrants thorough SAR studies. Moreover, **BE2623** was 3-fold less potent than **UK-5099** in inhibiting the respiration and was comparable to the 1*H*-pyrazol-4-yl derivative **BE1985** (IC_50_ = 0.638 µM). Elimination of the hydrogen acceptor (i.e., cyano group) as in **BE1975** abolished activity. Similarly, esterification of the carboxyl group as in **3a–f** and **BE1988** rendered the compound inactive.

We also calculated a set of molecular descriptors to evaluate the drug-likeness of the identified inhibitors using QikProp program [23]. Identified compounds possessed drug-like properties and complied with Lipinski’s Rule of Five. They were predicted to have good aqueous solubility, oral bioavailability, metabolic stability, and to be inactive in the central nervous system (Table 2). Human ether-a-go-go related gene (HERG) K+ channel blockers are potentially toxic; it is important to evaluate its inhibition as early as possible in drug discovery. The predicted IC_50_ values of our identified hits show that they are not predicted to cause cardiac toxicity. We will further test the in vitro absorption, distribution, metabolism, and excretion (ADME) of the best two lead compounds and optimize them accordingly for pharmacokinetics.

## 4. Discussion

The MPC complex is an attractive target for drug discovery, but the required properties for MPC inhibition and MPC complex composition are still unclear. The current study identified the putative substrate-binding cavity in the MPC dimer using homology modeling. Three amino acid residues Phe66 (MPC1) and Asn100 and Lys49 (MPC2) were validated by mutagenesis experiments to be important for substrate and/or inhibitor binding. While we showed that these mutant MPC1 or MPC2 proteins are individually expressed (Figure 3B), it is possible that these mutations affect the integrity of the MPC1/2 heterodimer. We do not suspect this for MPC1-F66A or MPC2-K49A, since there was still small, but significant, activation of BRET activity upon addition of MSDC-0602K or **UK-5099** in these mutants (Figure 3A). However, the magnitude of these increases was much less than in WT-expressing cells. Interestingly, several of these mutants altered the basal BRET activity in the absence of pyruvate or inhibitor. MPC1-N33A and MPC2-N100A increased BRET, while MPC2-K49A decreased BRET signal (Figure 3A). This suggests that these mutations may change the MPC1/2 heterodimer structure sufficiently to alter the physical distance between the C-terminally-linked NLuc BRET donor and mCherry BRET acceptor. Overall, this study further validated and provided a promising computational approach for further investigations on the prediction of MPC inhibitors with pharmacophore modeling.

Five novel, non-indole MPC inhibitors were identified and validated experimentally, with two of them showing activity in the low nanomolar range (Table 1 and Figure 6). These compounds inhibited mitochondrial respiration in a dose-dependent manner but did not reduce pyruvate respiration in MPC2−/− cardiac mitochondria indicating MPC-dependent inhibition. The novel inhibitors decreased pyruvate oxidation by MPC inhibition and not through reduction in PDH activity. Because of the pronounced activity of **BE1976** and **BE2617**, a more thorough characterization of its effects on pyruvate metabolism in cells and in vivo in future studies seems warranted. Additionally, a comprehensive structure-activity relationship study to optimize the pharmacokinetics of the novel inhibitors would provide important information for future drug discovery efforts aimed at inhibiting MPC activity.

The novel inhibitors described herein are structurally related to **UK-5099** because they bear the α-cyanoacrylic acid moiety. Although **UK-5099** was identified as an MPC inhibitor in 1975 [12], there were no efforts to optimize this lead compound until 2021 when Liu and co-workers developed novel **UK-5099** analogs as MPC inhibitors and used them topically to treat hair loss [11]. **UK-5099** and the novel identified hits possess a Michael acceptor unit. Michael acceptors were historically excluded from drug discovery due to fear of off-target effects and potential toxicity. Recent success in developing targeted Michael acceptor drugs and the surge in number of FDA approved drugs and clinical candidates containing Michael acceptors, renewed the interest in this important category of compounds [28]. The identification of these new core scaffolds increases the chemical space around this scaffold and provides opportunity for drug design.

Inhibition of the MPC complex may be beneficial in a variety of chronic and progressive diseases. In some types of cancers, loss of MPC function has been related to cancer growth due to the role it plays in Warburg effect. Despite this, MPC inhibition has been established as a mechanism of action for lonidamine, an anti-tumor drug used to sensitize tumors to chemotherapy and radiotherapy [29], suggesting that mitochondrial pyruvate metabolism is important for cancer cell proliferation. MPC is also an attractive therapeutic target to treat neurodegenerative diseases linked to excitotoxicity [30]. Indeed, MPC inhibition protected primary cortical neurons from excitotoxic death without compromising energy metabolism, probably because of the metabolic flexibility of neurons [30]. Regulation of blood glucose concentrations and efficient hepatic gluconeogenesis require uptake and metabolism of pyruvate by mitochondria. Inhibition of the MPC in hepatocytes attenuates hyperglycemia in preclinical diabetes models; efficacy has also been demonstrated in clinical trials [5,6,31]. Moreover, growing scientific evidence suggests that targeting MPC is an effective strategy for the treatment of nonalcoholic fatty liver disease (NAFLD) and nonalcoholic steatohepatitis (NASH) [31,32]. A TZD MPC inhibitor, MSDC-0602, was shown to prevent liver fibrosis and reduce hepatic stellate cell activation in vitro [32]. Deletion of liver MPC2 in mice fed with trans-fatty acids rich diet protected them from developing NASH and reduced stellate cell activation [32]. Future studies will evaluate the effects of these novel inhibitors on relevant metabolic and pathogenic endpoints to determine their efficacy for treating diseases of interest.

## 5. Conclusions

Herein, we created a homology model of the MPC structure based upon similar protein structures for the bacterial SemiSWEET crystal structures. To validate this model structure, we performed site-directed mutagenesis and identified several amino acid residues that prevent the ability of a BRET based sensor (also newly developed in these studies) to bind pyruvate or known MPC inhibitors. We next generated a pharmacophore model that represent chemical and geometrical features of known MPC inhibitors and performed pharmacophore-based virtual compound screen of one million compounds library. We tested several of these screened chemicals in both BRET assays and in isolated mitochondrial respiration studies to validate MPC inhibition and determine relative IC_50_ concentrations for these novel inhibitors. We also describe that these inhibitors do not decrease pyruvate respiration in MPC−/− mitochondria, nor do they alter phosphorylation of pyruvate dehydrogenase (a proxy of PDH activity); thus, they appear to specifically inhibit the MPC at low nanomolar concentrations. Lastly, we performed a predicted absorption, distribution, metabolism, and excretion (ADME) analysis on these compounds, which suggests good solubility and bioavailability. In future studies, we will advance these compounds to preclinical models of metabolic disease. But we believe this current work is a significant step forward in gaining understanding of the MPC structure and development of novel inhibitor compounds.

## Data Availability

The datasets generated and/or analyzed in this study are available on reasonable request.

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
