# Peer review of "Identification of Novel Mitochondrial Pyruvate Carrier Inhibitors by Homology Modeling and Pharmacophore-Based Virtual Screening"

_biomedicines, 2022, doi:10.3390/biomedicines10020365_

Round 1

Reviewer 1 Report

The authors present their work, in which they have carried out the first steps of a drug discovery campaign to identify novel inhibitors of the mitochondrial pyruvate carrier (MPC1/MPC2 heterodimer), with the aim of potentially improving on the efficiency of existing inhibitors. In addtion, they have constructed a homology-based model of the MPC dimer, based on which they identified and validated residues that likely play a role in substrate and inhibitor transport and thus form part of the substrate-binding site of the transporter. In their campaign, the authors have performed a pharmacophore-based virtual screen of a large compound library, and have successfully synthesized novel inhibitors that despite sharing a similar scaffold to the original UK-5099 compound, could significantly improve on its affinity. Software prediction suggests that the novel compounds bear drug-like properties.

MPC is a currently active field of research due to its involvement in crucial physiological processes and diseases, as the authors summarize in their introductory section. Improved and specific inhibitors especially in the nanomolar range can be valuable tools for both basic research and the development of therapeutics at a later stage. Nevertheless, unfortunately the specificity of the presented novel compounds has not been experimentally tested by the authors, thus this remains an open question. Novelty of these compounds is also slightly diminished due to their similarity to the parent compound UK-5099, however, notably, the authors have succeeded to improve the IC50 of the parent compound by one magnitude.

In addition, the authors construct a homology-based structural model of the MPC1/MPC2 heterodimer, which they analyze for possible substrate-binding residues. While not strictly necessary for supporting their results, the work would have benefitted from computational docking analysis of the developed compounds and pyrophosphate (substrate) to fully exploit the potential of a structural model, and provide a structural explanation of the observed structure-activity relationships. Nevertheless, the authors still identified residues that, based on their results, are compatible with the pharmacophore model used and thus constitute a plausible substrate- and inhibitor-binding site.

The overall quality of the results and their interpretation is high and scientifically sound. However, certain modifications are necessary to the manuscript to improve the presentation of the results, and in my view a few questions should be clarified in the manuscript.
I list them below:
- The weights used in eq. (1) are introduced but their source is not defined, according to the documentation of the Phase software, these are user-adjustable parameters, and if so, their values and the logic of determining the values used should be discussed in the manuscript, as they affect the results of virtual screening.
- Statistical significance of groups compared in Figure 3 should be calculared appropriately and presented.
- It should be discussed whether the introduction of mutations in MPC1 and MPC2 affect the integrity of the MPC1/MPC2 heterodimer, especially in light of their effect on decreasing the control BRET signal in some cases (e.g. Figure 3A, mutant MPC2-K49A DMSO control vs WT). It could be that decreased ligand binding was caused by the rupture of the MPC1/MPC2 heterodimer due to the mutation. Increase of baseline BRET signal due to mutations should also be interesting to discuss (e.g. Figure 3A, MPC2-N100A vs WT).
- It should be clarified why the UK-5099 compound in Table 1 has a PhaseScreenScore of 2.66, while in line 316 the authors claim that "UK-5099 should have a maximal score of 3".
- The authors could elaborate how the 7 compounds were chosen for further studies. Maybe an analysis of the distribution of PhaseScreenScore values would help. Was a cut-off applied? Were the "top n" compounds taken? Were the compounds selected based on their similarity to the parent compound UK-5099?
- In Figure 3A, the shade of the legend does not show properly which graphs are WT and which are the mutant, this should be corrected.
- Figure 6D seems to contain red wavy underlined text, probably due to spell checking, this should be removed.
- The sentence in line 412 ("UK-5099 and novel identified hits possess Michael acceptor unit and historically similar scaffolds were excluded from drug discovery due to the fear of off-target effects and potential toxicity.") is malformed and should be corrected or rephrased.

With these comments in mind, I would suggest a revision of the manuscript accordingly.

Author Response

MPC is a currently active field of research due to its involvement in crucial physiological processes and diseases, as the authors summarize in their introductory section. Improved and specific inhibitors especially in the nanomolar range can be valuable tools for both basic research and the development of therapeutics at a later stage. Nevertheless, unfortunately the specificity of the presented novel compounds has not been experimentally tested by the authors, thus this remains an open question. Novelty of these compounds is also slightly diminished due to their similarity to the parent compound UK-5099, however, notably, the authors have succeeded to improve the IC50 of the parent compound by one magnitude.

In addition, the authors construct a homology-based structural model of the MPC1/MPC2 heterodimer, which they analyze for possible substrate-binding residues. While not strictly necessary for supporting their results, the work would have benefitted from computational docking analysis of the developed compounds and pyrophosphate (substrate) to fully exploit the potential of a structural model, and provide a structural explanation of the observed structure-activity relationships. Nevertheless, the authors still identified residues that, based on their results, are compatible with the pharmacophore model used and thus constitute a plausible substrate- and inhibitor-binding site.

The overall quality of the results and their interpretation is high and scientifically sound. However, certain modifications are necessary to the manuscript to improve the presentation of the results, and in my view a few questions should be clarified in the manuscript.

Thank you very much for your enthusiasm for this manuscript. These are all good suggestions, and we believe the revisions have significantly improved the manuscript. Our responses to these specific critiques are listed below in blue text.

I list them below:
- The weights used in eq. (1) are introduced but their source is not defined, according to the documentation of the Phase software, these are user-adjustable parameters, and if so, their values and the logic of determining the values used should be discussed in the manuscript, as they affect the results of virtual screening.

Thank you for bringing this to our attention. The user adjustable parameters were kept at their default values. We have updated this information and corresponding parameter values in the methods section of the manuscript.

- Statistical significance of groups compared in Figure 3 should be calculated appropriately and presented.

We apologize for omitting the statistical comparisons in this figure. We have now added the symbols to indicate statistical significance based on Two-way ANOVA with Tukey’s correction for multiple comparisons.

- It should be discussed whether the introduction of mutations in MPC1 and MPC2 affect the integrity of the MPC1/MPC2 heterodimer, especially in light of their effect on decreasing the control BRET signal in some cases (e.g. Figure 3A, mutant MPC2-K49A DMSO control vs WT). It could be that decreased ligand binding was caused by the rupture of the MPC1/MPC2 heterodimer due to the mutation. Increase of baseline BRET signal due to mutations should also be interesting to discuss (e.g. Figure 3A, MPC2-N100A vs WT).

Thank you for suggesting these, which are very good points of discussion. We have added discussion sentences to further describe these points, in that the mutations may affect the heterodimer formation, and that some of the mutations altered the baseline (DMSO vehicle) BRET activity, and hypothesized why that may be.

- It should be clarified why the UK-5099 compound in Table 1 has a PhaseScreenScore of 2.66, while in line 316 the authors claim that "UK-5099 should have a maximal score of 3".

Thank you for raising this concern. In line 316, we stated that “The reference ligand which matches exactly has a perfect score of 3”. The reason that UK-5099 has a PhaseScreenScore of 2.66 instead of 3, is because the generated pharmacophore model doesn’t account for the phenyl ring attached to the indole nitrogen (which we believe is not critical for binding) and therefore the pharmacophore model doesn’t exactly match UK-5099. We removed the sentence “The reference ligand which matches exactly has a perfect score of 3" to avoid confusion.

- The authors could elaborate how the 7 compounds were chosen for further studies. Maybe an analysis of the distribution of PhaseScreenScore values would help. Was a cut-off applied? Were the "top n" compounds taken? Were the compounds selected based on their similarity to the parent compound UK-5099?

The top 7 compounds were selected based on their PhaseScreenScore. This information was added to the results section of the manuscript.

- In Figure 3A, the shade of the legend does not show properly which graphs are WT and which are the mutant, this should be corrected.

Thank you for catching this labelling. We have reformatted this figure, and now the genotypes grouped together and also labelled below the graphs. This helps with the labelling, and also simplified our indicating of the statistical comparisons.

- Figure 6D seems to contain red wavy underlined text, probably due to spell checking, this should be removed.

Again, thank you for catching this. We have removed the spell check indicators here.

- The sentence in line 412 ("UK-5099 and novel identified hits possess Michael acceptor unit and historically similar scaffolds were excluded from drug discovery due to the fear of off-target effects and potential toxicity.") is malformed and should be corrected or rephrased.

We agree this sentence was confusing. We have revised it.

With these comments in mind, I would suggest a revision of the manuscript accordingly.

Thank you, we hope you find these responses and corresponding revisions as significant improvements to the manuscript.

Reviewer 2 Report

This study has revealed a few MPC inhibitors with novel scaffold and the authors have done excellent experimental design.
Homology modeling and site-directed mutagenesis were the key for further experiments including pharmacophore modeling.
In Figure 1 and Figure 4, the bioactivity values of chemicals can be included for easy access to information.
On what basis the seven compounds were selected for biological testing? Only based on Phase screen score or any other method such as molecular docking was used to confirm or rescore?  
In Discussion, you can cite the figure and table in the following sentence "Four novel, non-indole MPC inhibitors were identified and validated experimentally with two of them showing activity in the nanomolar range".
Overall, I find this study useful and presented with enough evidences.

Author Response

This study has revealed a few MPC inhibitors with novel scaffold and the authors have done excellent experimental design.
Homology modeling and site-directed mutagenesis were the key for further experiments including pharmacophore modeling.

Thank you very much for the kind words and enthusiasm regarding our manuscript. Thank you for these good suggestions which certainly helped us improve the manuscript. Our responses to individual critiques can be found below in blue text:

In Figure 1 and Figure 4, the bioactivity values of chemicals can be included for easy access to information.

Thank you for this suggestion. We did not study all of the compounds in Figure 1, which have been previously studied in detail as MPC inhibitors. We simply provide their structures as comparisons of the two main historical MPC inhibitor classes (cyanocinnamates or thiazolidinediones). We now list the determined relative IC50s for the compounds shown in Figure 5.

On what basis the seven compounds were selected for biological testing? Only based on Phase screen score or any other method such as molecular docking was used to confirm or rescore?  

Thank you for your question. We used insights from homology modeling and mutagenesis data which suggested the molecular elements of binding and we used this information for generation of the pharmacophore model. We relied only on the PhaseScreenScore to rank the virtually screened compounds based on the alignment of their chemical features with the generated pharmacophore hypothesis. We selected the top 7 scored compounds which ensures they have same chemical features required by the developed pharmacophore model for binding. We have added this information to the manuscript results description.

In Discussion, you can cite the figure and table in the following sentence "Four novel, non-indole MPC inhibitors were identified and validated experimentally with two of them showing activity in the nanomolar range".

Thank you, we now cite Table 1 and Figure 6 after this sentence.

Overall, I find this study useful and presented with enough evidence.

Thank you very much!
